# ResWorld: Temporal Residual World Model for End-to-End Autonomous Driving

**Jinqing Zhang[1], Zehua Fu[4], Zelin Xu[3], Wenying Dai[3], Qingjie Liu[1,2,4*], Yunhong Wang[1,4]**

[1]State Key Laboratory of Virtual Reality Technology and Systems, Beihang University, Beijing, China
[2]Zhongguancun Laboratory, Beijing, China
[3]Beijing Jingwei Hirain Technologies Co., Inc.
[4]Hangzhou Innovation Institute, Beihang University, Hangzhou, China

## Abstract

The comprehensive understanding capabilities of world models for driving scenarios have significantly improved the planning accuracy of end-to-end autonomous driving frameworks. However, the redundant modeling of static regions and the lack of deep interaction with trajectories hinder world models from exerting their full effectiveness. In this paper, we propose Temporal Residual World Model (TR-World), which focuses on dynamic object modeling. By calculating the temporal residuals of scene representations, the information of dynamic objects can be extracted without relying on detection and tracking. TR-World takes only temporal residuals as input, thus predicting the future spatial distribution of dynamic objects more precisely. By combining the prediction with the static object information contained in the current BEV features, accurate future BEV features can be obtained. Furthermore, we propose Future-Guided Trajectory Refinement (FGTR) module, which conducts interaction between prior trajectories (predicted from the current scene representation) and the future BEV features. This module can not only utilize future road conditions to refine trajectories, but also provides sparse spatial-temporal supervision on future BEV features to prevent world model collapse. Comprehensive experiments conducted on the nuScenes and NAVSIM datasets demonstrate that our method, namely ResWorld, achieves state-of-the-art planning performance. The code is available at https://github.com/mengtan00/ResWorld.git.

## 1 Introduction

End-to-end autonomous driving framework has emerged as an important research direction in recent years, presenting a cost-effective and highly scalable solution for autonomous driving applications. The traditional autonomous driving systems generally perform environmental perception at first, including 3D object detection (Huang et al., 2021; Wang et al., 2023; Zhang et al., 2023; 2025), map segmentation (Li et al., 2022a; Liao et al., 2022; 2024; Yuan et al., 2024b) and semantic occupancy prediction (Ma et al., 2024; Yu et al., 2023; Zhang et al., 2024), among others. Subsequently, the multiple perception results are integrated through rule-based methods (Bouchard et al., 2022; Treiber et al., 2000) or independent DNN models (Huang et al., 2023; Cheng et al., 2024a;b) to generate the future trajectory of the ego vehicle. In contrast, end-to-end autonomous driving frameworks (Hu et al., 2023; Jiang et al., 2023; Sima et al., 2023; Zheng et al., 2024b; Sun et al., 2024; Weng et al., 2024; Hu et al., 2022a; Guo et al., 2024) integrate the multiple tasks into a single model. Such a design not only reduces information loss between raw data and the final planning results but also enables collaborative optimization across various modules, thus exhibiting stronger adaptability to complex scenarios.

Recently, due to the high annotation cost required for training multiple perception and prediction modules, several end-to-end autonomous driving approaches (Li & Cui, 2025; Li et al., 2025; Zheng et al., 2024a; Yang et al., 2025; Zheng et al., 2025) have adopted world models to replace these auxiliary task modules. As shown in Fig 1a, by treating future scene prediction as a proxy task, world

---

*Corresponding author: qingjie.liu@buaa.edu.cn

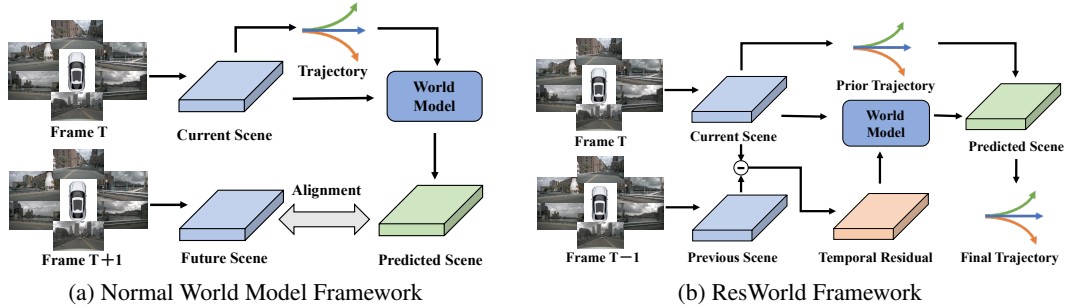

(a) Normal World Model Framework          (b) ResWorld Framework

Figure 1: **Comparison Between Normal World Model Framework and ResWorld Framework.** Different from the normal world models that model the entire scene and implicitly optimize trajectories, Resworld uses the temporal residuals of the scene representations to represent dynamic objects for precise modeling. Meanwhile, the prior trajectories are corrected through explicit interaction with the predicted future BEV feature.

model frameworks can effectively enhance the model's ability to understand and model driving scenes, thereby improving the planning accuracy. However, most information in the scene representations belongs to static objects such as the ground and buildings, which can be directly retained in future scenarios without the need for redundant modeling. In contrast, dynamic objects such as vehicles and pedestrians require more precise modeling, yet they are difficult to identify from the environment without relying on perception tasks. Furthermore, current methods lack deep interaction between trajectories and the future scene representations predicted by the world model.

To address these issues, we propose Temporal Residual World Model (TR-World) as shown in Fig 1b, which can precisely model the dynamic objects and predict accurate future scene representations. First, we shift BEV features at different timestamps into the current BEV coordinate system and use the same spatial attention mask to extract their sparse scene queries. Subsequently, we subtract the scene queries of adjacent timestamps to obtain the temporal residuals of scene queries. The temporal residuals represent the changes in the same position across different timestamps, thus standing for the dynamic objects in the scene. When predicting future BEV features, the current BEV coordinate system is still adopted. This allows the current BEV features to depict the future distribution of static objects, thereby avoiding redundant modeling of static objects. TR-World only processes the temporal residuals and maps the predicted future spatial distribution of dynamic objects onto the current BEV features, thereby obtaining accurate predictions of future BEV features.

Furthermore, to make full use of the predicted future BEV features, we propose Future-Guided Trajectory Refinement (FGTR) module. We first employ a series of waypoint queries to represent the ego vehicle's future trajectory, where each query corresponds to the ego vehicle's position at a specific future timestamp. After decoding the prior trajectory from waypoint queries, this trajectory acts as a set of reference points to guide the interaction between waypoint queries and future BEV features. This operation can effectively verify whether the prior trajectory will collide with other objects or deviate from the drivable area, thereby correcting the prior trajectory and improving the planning performance. FGTR additionally applies sparse spatial-temporal supervision on future BEV features, which can effectively alleviate world model collapse. It is worth noting that supervising future BEV features with ground truth at any future timestamps will lose the spatial distribution of dynamic objects at other timestamps. Therefore, not applying supervision can instead allow the model to independently optimize the future BEV features and retain the most important information.

We integrate the proposed components into a novel end-to-end autonomous driving model, namely ResWorld. Experiments conducted on nuScenes and NAVSIM benchmarks indicate that ResWorld achieves state-of-the-art planning accuracy. Our contributions can be summarized as follows:

- We use the current BEV coordinate system to represent the future BEV representations predicted by the world model, eliminating the need for redundant modeling of static objects.
- We utilize the temporal residuals of scene representations to extract information about dynamic objects without relying on auxiliary tasks. Temporal residuals are processed by Temporal Residual World Model to predict dynamic objects' future spatial distribution.

- We propose Future-Guided Trajectory Refinement module, which applies interaction between prior trajectory and future BEV features to improve the planning accuracy and prevent world model collapse.
- ResWorld achieves state-of-the-art results on nuScenes and NAVSIM benchmarks, demonstrating the effectiveness of our proposed framework.

## 2   RELATED WORKS

### 2.1   END-TO-END AUTONOMOUS DRIVING

Nowadays, end-to-end autonomous driving approaches are gaining increasing attention for cost-effectiveness and high scalability. These methods generally adopt an integrated model that predicts trajectories from the input raw sensor data, achieving state-of-the-art trajectory prediction performance. ST-P3 (Hu et al., 2022b) obtains future ego-vehicle movements by progressively utilizing map perception module, BEV occupancy module, and planning module. UniAD (Hu et al., 2023) enhances the robustness of the system by further adopting supplementary detection, tracking, and motion prediction modules. VAD (Jiang et al., 2023) represents object movements and lane lines using vectors, thereby reducing the total computation of the model. PARA-Drive (Weng et al., 2024) comprehensively explores the design space of modular perception and prediction task stacks in autonomous driving. OccNet (Sima et al., 2023) introduces occupancy prediction to construct detailed 3D scene representations for planning. VADv2 (Chen et al., 2024) predicts multiple action candidates and samples one action as the planning result. GenAD (Zheng et al., 2024b) adopts the generative model for trajectory generation, jointly optimizing motion and planning heads. UAD (Guo et al., 2024) utilizes the angular object mask as the scene representation to avoid collision. DiffusionDrive (Liao et al., 2025) applies diffusion models to boost trajectories' diversity and robustness in complex scenarios. However, these methods rely on fine-grained annotations to train auxiliary task modules, which restricts their ability to utilize large-scale raw data.

### 2.2   WORLD MODEL FOR END-TO-END AUTONOMOUS DRIVING

World models have demonstrated excellent spatial understanding and modeling capabilities, which are leveraged by some end-to-end autonomous driving models to replace auxiliary tasks. OccWorld (Zheng et al., 2024a) adopts unified occupancy-centric world modeling, enhancing spatial-temporal scene understanding for robust planning. Drive-WM (Wang et al., 2024) generates high-quality driving videos through joint spatial-temporal modeling, thereby improving the model's planning accuracy. SSR (Li & Cui, 2025) converts the dense BEV feature into sparse scene queries and utilizes the world model to enhance the scene understanding. LAW (Li et al., 2024a) adopts the latent world model framework and carries out experiments under perception-free and perception-based settings. World4Drive (Zheng et al., 2025) generates multi-modal trajectories and utilizes the world model to select the most appropriate one. However, these world models tend to perform redundant modeling on static objects, while their modeling of dynamic objects remains insufficient. Additionally, the absence of deep interaction between trajectories and future scene representations hinders world models from exerting their full effectiveness.

## 3   METHOD

### 3.1   PRIOR TRAJECTORY PREDICTION

To extract the temporal residuals of BEV features, it is necessary for BEV features to have high geometric quality, which facilitates the spatial alignment of BEV features across different timestamps. Therefore, we choose GeoBEV (Zhang et al., 2025) as the base of the model, which can efficiently generate BEV features with high geometric quality. As shown in Fig 2, the multi-view images for each timestamp are converted to BEV features, thus obtaining $\{\mathbf{B}_t, \mathbf{B}_{t-1}, \ldots, \mathbf{B}_{t-k}\}$. $\mathbf{B}_t \in \mathbb{R}^{C \times H \times W}$ is the BEV feature of the current timestamp, where $C, H, W$ are the channel, height and width dimensions, and $k$ is the number of past timestamps. Following BEVDet4D (Huang & Huang, 2022), $\{\mathbf{B}_{t-1}, \ldots, \mathbf{B}_{t-k}\}$ are all transformed into the coordinate system of $\mathbf{B}_t$ and fused by

$$\mathbf{B}_{fuse} = \text{Conv}(\text{Concat}(\mathbf{B}_t, \mathbf{B}_{t-1}, \ldots, \mathbf{B}_{t-k})) \qquad (1)$$

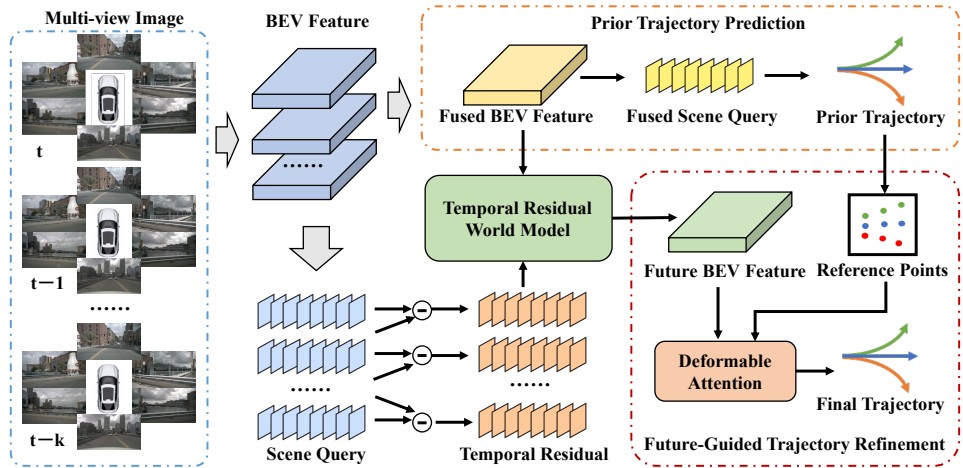

Figure 2: **Overall Framework of ResWorld.** Multi-view images at different timestamps are converted into BEV features, which are used to predict prior trajectories. On the other hand, BEV features are used to calculate temporal residuals, which are then processed by the Temporal Residual World Model to predict the future distribution of dynamic objects. Future-Guided Trajectory Refinement module further utilizes the predicted future BEV features to refine the planning results.

We use the planning module of SSR (Li & Cui, 2025) to perform perception-free planning. The dense $\mathbf{B}_{fuse}$ is first processed by a TokenLearner module (Ryoo et al., 2021) to obtain $N_s$ sparse scene queries $\mathbf{S}_{fuse} \in \mathbb{R}^{N_s \times C}$, which can be formulated by

$$\mathbf{S}_{fuse} = \text{TokenLearner}(\mathbf{B}_{fuse}) = \text{AvgPool}(\text{SA}(\mathbf{B}_{fuse}) \odot \mathbf{B}_{fuse}) \tag{2}$$

where SA denotes the generation of the spatial attention map and AvgPool denotes the global average pooling operation. $\mathbf{S}_{fuse}$ is operated by self-attention for further information extraction:

$$\mathbf{S}_{fuse} = \text{SelfAttention}(\mathbf{S}_{fuse}) \tag{3}$$

We use a set of waypoint queries $\mathbf{W} \in \mathbb{R}^{N_t \times C}$ to represent the ego vehicle's future status, where $N_t$ denotes the number of future timestamps to be predicted. After the cross attention operation between $\mathbf{W}$ and $\mathbf{S}_{fuse}$, the prior trajectories can be decoded by a multi-layer perceptron (MLP) as:

$$\mathbf{T}_{prior} = \text{MLP}(\text{CrossAttention}(\mathbf{W}, \mathbf{S}_{fuse}, \mathbf{S}_{fuse})) \tag{4}$$

where each row in $\mathbf{T}_{prior} \in \mathbb{R}^{N_t \times 2}$ represents the ego vehicle's coordinates at a future timestamp.

### 3.2 TEMPORAL RESIDUAL EXTRACTION

Since $\{\mathbf{B}_t, \mathbf{B}_{t-1}, \ldots, \mathbf{B}_{t-k}\}$ share the same coordinate system of $\mathbf{B}_t$, they represent the scene information of the same scene at different timestamps. By calculating their residuals, information about dynamic objects in the scene can be extracted.

Given that $\mathbf{B}_{fuse}$ carries the spatial information across different timestamps, it can be utilized to predict a spatial attention map that emphasizes the regions with dynamic objects. For each timestamp $i$, $\mathbf{B}_i$ is weighted by this spatial attention map to extract the sparse scene queries formulated as

$$\mathbf{S}_i = \text{AvgPool}(\text{SA}(\mathbf{B}_{fuse}) \odot \mathbf{B}_i) \tag{5}$$

After obtaining $\{\mathbf{S}_t, \mathbf{S}_{t-1}, \ldots, \mathbf{S}_{t-k}\}$, a set of temporal residuals $\{\mathbf{R}_t, \mathbf{R}_{t-1}, \ldots, \mathbf{R}_{t-k+1}\}$ is calculated by subtracting scene queries of the previous timestamp as shown in Fig 2.

### 3.3 TEMPORAL RESIDUAL WORLD MODEL

Previous world models used for end-to-end autonomous driving do not distinguish between dynamic objects and static objects in the scene and devote the same effort to predicting their future spatial

Figure 3: **Structure of Temporal Residual World Model**

distribution. However, if the coordinate system of $\mathbf{B}_t$ is still adopted when predicting future BEV features, the spatial distribution of static objects can be regarded as unchanged. As a result, $\mathbf{B}_{fuse}$ can serve as the appropriate future representation of static objects, eliminating the need for additional modeling. In addition, the understanding of static objects is already accomplished during the prediction of prior trajectories, and the world model is not required to participate in this process.

To avoid redundant modeling of static objects and make the world model focus more on dynamic objects, we propose the Temporal Residual World Model (TR-World) as shown in Fig 3. TR-World takes only temporal residuals as input to predict the future spatial distribution of dynamic objects. Specifically, each temporal residual $\mathbf{R}_i$ undergoes information extraction via self-attention operations, followed by accumulation across timestamps to obtain a future representation of dynamic objects $\hat{\mathbf{R}} \in \mathbb{R}^{N_s \times C}$. This process can be formulated as

$$\hat{\mathbf{R}} = \sum_{i=t-k+1}^{t} \text{SelfAttention}(\mathbf{R}_i) \tag{6}$$

$\hat{\mathbf{R}}$ needs to be presented on BEV features to restore the future spatial distribution of the dynamic objects accurately. We adopt TokenFuser (Ryoo et al., 2021), the inverse transformation of TokenLearner, to expand $\hat{\mathbf{R}}$ on the base of $\mathbf{B}_{fuse}$ by

$$\mathbf{B}_{future} = \text{TokenFuser}(\hat{\mathbf{R}}, \mathbf{B}_{fuse}) + \mathbf{B}_{fuse} = \text{MLP}(\mathbf{B}_{fuse}) \otimes \hat{\mathbf{R}} + \mathbf{B}_{fuse} \tag{7}$$

where MLP maps $\mathbf{B}_{fuse}$ to $\mathbb{R}^{N_s \times H \times W}$ and $\otimes$ denotes the combination of matrix transposition and multiplication, which outputs the prediction of future BEV features $\mathbf{B}_{future} \in \mathbb{R}^{C \times H \times W}$.

### 3.4 FUTURE-GUIDED TRAJECTORY REFINEMENT

Existing end-to-end autonomous driving methods generally utilize the world model to optimize planning performance in an indirect manner. Specifically, by treating the prediction of future scene representations as a proxy task, the model's overall ability to understand autonomous driving scenarios can be enhanced. However, the predicted future scene representations could serve as valuable references for trajectory planning, yet they have not been effectively utilized to date. On the other hand, when future scene representations lack supervision from any auxiliary tasks, it is challenging to prevent the world model from collapsing, which means the model tends to map diverse driving scenes to identical scene representations.

To address the above issues, we have designed the Future-Guided Trajectory Refinement (FGTR) module. This module simply applies Deformable Attention Operation between waypoint queries $\mathbf{W}$ and future BEV features $\mathbf{B}_{future}$, while $\mathbf{T}_{prior}$ serves as the reference points on $\mathbf{B}_{future}$ as shown in Fig 2. Subsequently, the final trajectory $\mathbf{T}_{final}$ is decoded by MLP, which can be formulated by

$$\mathbf{W} = \text{DeformAttention}(\mathbf{W}, \mathbf{B}_{future}, \mathbf{T}_{prior}) \tag{8}$$

$$\mathbf{T}_{final} = \text{MLP}(\mathbf{W}) \tag{9}$$

Since each query in $\mathbf{W}$ represents the ego vehicle's status at a future timestamp, FGTR module can collect the future environmental information around the ego vehicle from $\mathbf{B}_{future}$ based on $\mathbf{T}_{prior}$. This information can be used to check whether the ego vehicle will collide with other objects or drive out of the drivable area, and thus correct $\mathbf{T}_{prior}$ promptly. This not only makes full use of $\mathbf{B}_{future}$ but also provides sparse spatial-temporal supervision for it. While reference point coordinates provide

Table 1: **Comparison of state-of-the-art methods on the nuScenes dataset**. ∗ denotes the metrics evaluated using the official models and code. ◇ denotes using ego status in the planning module following BEVPlanner++ (Li et al., 2024c). ‡ denotes the AVG metric calculated in the same way as VAD (Jiang et al., 2023).

| Method | Auxiliary Task | L2 (m) ↓ | | | | Collision Rate (%) ↓ | | | |
|---|---|---|---|---|---|---|---|---|---|
| | | 1s | 2s | 3s | Avg | 1s | 2s | 3s | Avg |
| ST-P3 (Hu et al., 2022b) | Det&Map | 1.72 | 3.26 | 4.86 | 3.28 | 0.44 | 1.08 | 3.01 | 1.51 |
| UniAD (Hu et al., 2023) | Det&Track&Map&Motion&Occ | 0.48 | 0.96 | 1.65 | 1.03 | 0.05 | 0.17 | 0.71 | 0.31 |
| OccNet (Sima et al., 2023) | Det&Map&Occ | 1.29 | 2.13 | 2.99 | 2.14 | 0.21 | 0.59 | 1.37 | 0.72 |
| PARA-Drive (Weng et al., 2024) | Det&Track&Map&Motion&Occ | 0.40 | 0.77 | 1.31 | 0.83 | 0.07 | 0.25 | 0.60 | 0.30 |
| GenAD (Zheng et al., 2024b) | Det&Map&Motion | 0.36 | 0.83 | 1.55 | 0.91 | 0.06 | 0.23 | 1.00 | 0.43 |
| SSR∗ (Li & Cui, 2025) | None | 0.25 | 0.64 | 1.33 | 0.74 | 0.08 | 0.12 | 0.72 | 0.31 |
| ResWorld | None | 0.22 | 0.56 | 1.17 | 0.65 | **0.02** | **0.04** | 0.64 | 0.23 |
| ResWorld◇ | None | **0.19** | **0.50** | **1.08** | **0.59** | **0.02** | 0.06 | **0.43** | **0.17** |
| ST-P3‡ (Hu et al., 2022b) | Det&Map | 1.33 | 2.11 | 2.90 | 2.11 | 0.23 | 0.62 | 1.27 | 0.71 |
| UniAD‡ (Hu et al., 2023) | Det&Track&Map&Motion&Occ | 0.44 | 0.67 | 0.96 | 0.69 | 0.04 | 0.08 | 0.23 | 0.12 |
| VAD‡ (Jiang et al., 2023) | Det&Map&Motion | 0.41 | 0.70 | 1.05 | 0.72 | 0.07 | 0.17 | 0.41 | 0.22 |
| BEV-Planner++◇‡ (Li et al., 2024c) | None | 0.16 | 0.32 | 0.57 | 0.35 | 0.00 | 0.29 | 0.73 | 0.34 |
| PARA-Drive‡ (Weng et al., 2024) | Det&Track&Map&Motion&Occ | 0.25 | 0.46 | 0.74 | 0.48 | 0.14 | 0.23 | 0.39 | 0.25 |
| LAW‡ (Li et al., 2025) | None | 0.26 | 0.57 | 1.01 | 0.61 | 0.14 | 0.21 | 0.54 | 0.30 |
| LAW‡ (Li et al., 2025) | Det&Map&Motion | 0.24 | 0.46 | 0.76 | 0.49 | 0.08 | 0.10 | 0.39 | 0.19 |
| GenAD‡ (Zheng et al., 2024b) | Det&Map&Motion | 0.28 | 0.49 | 0.78 | 0.52 | 0.08 | 0.14 | 0.34 | 0.19 |
| SparseDrive‡ (Sun et al., 2024) | Det&Track&Map&Motion | 0.29 | 0.58 | 0.96 | 0.61 | 0.01 | 0.05 | 0.18 | 0.08 |
| Drive-OccWorld‡ (Yang et al., 2025) | Occ | 0.25 | 0.44 | 0.72 | 0.47 | 0.03 | 0.08 | 0.22 | 0.11 |
| SSR∗‡ (Li & Cui, 2025) | None | 0.19 | 0.36 | 0.62 | 0.39 | 0.10 | 0.10 | 0.24 | 0.15 |
| MomAD‡ (Song et al., 2025) | Det&Track&Map&Motion | 0.31 | 0.57 | 0.91 | 0.60 | 0.01 | 0.05 | 0.22 | 0.09 |
| DiffusionDrive‡ (Liao et al., 2025) | Det&Track&Map&Motion | 0.27 | 0.54 | 0.90 | 0.57 | 0.03 | 0.05 | 0.16 | 0.08 |
| ResWorld‡ | None | 0.17 | 0.32 | 0.55 | 0.35 | **0.01** | **0.02** | 0.16 | 0.07 |
| ResWorld◇‡ | None | **0.14** | **0.27** | **0.49** | **0.30** | **0.01** | 0.03 | **0.14** | **0.06** |

spatial supervision, the different timestamps represented by $\mathbf{W}$ offer temporal supervision. $\mathbf{B}_{future}$ is encouraged to accurately represent cross-temporal spatial information such as the future positions of dynamic objects, thus preventing the world model from collapsing.

## 3.5 Loss

During training, we only adopts the L1 loss for $\mathbf{T}_{prior}$ and $\mathbf{T}_{final}$, which can be expressed as

$$\mathcal{L} = \text{L1}(\mathbf{T}_{prior}, \mathbf{T}_{GT}) + \text{L1}(\mathbf{T}_{final}, \mathbf{T}_{GT}) \tag{10}$$

where $\mathbf{T}_{GT}$ denotes the ground truth trajectory of ego vehicle. Unlike general world models, we do not utilize real future data to generate the label for supervising $\mathbf{B}_{future}$. This approach enables $\mathbf{B}_{future}$ to preserve the spatial distribution of dynamic objects across multiple future timestamps, rather than being limited to a specific timestamp. Experiments confirm that not supervising $\mathbf{B}_{future}$ enables higher planning performance.

## 4 Experiment Resuls

### 4.1 Dataset and Metric

**nuScenes** We conduct the open-loop evaluation of ResWorld on nuScenes (Caesar et al., 2020), the commonly used autonomous driving dataset. Consistent with previous works (Li & Cui, 2025; Li et al., 2025), we use displacement error and collision rate (CR) as metrics to evaluate the accuracy of trajectory prediction. The displacement error is represented by the L2 error between the predicted trajectory and the ground truth trajectory, which indicates the degree of deviation between the planning model and the experts. The collision rate quantifies the percentage of cases involving collisions with other objects when executing the predicted trajectory, reflecting the safety of the

Table 2: **Comparison of state-of-the-art methods on the NAVSIM navtest split**. $\star$ denotes the utilization of historical frame to obtain the temporal residual of the scene represenation.

| Method | Auxiliary Task | NC ↑ | DAC ↑ | TTC ↑ | Comf. ↑ | EP ↑ | PDMS ↑ |
|---|---|---|---|---|---|---|---|
| LAW (Li et al., 2025) | None | 96.4 | 95.4 | 88.7 | 99.9 | 81.7 | 84.6 |
| World4Drive (Zheng et al., 2025) | None | 97.4 | 94.3 | 92.8 | **100** | 79.9 | 85.1 |
| ResWorld | None | 98.1 | 95.6 | 94.3 | **100** | 81.8 | 87.3 |
| UniAD (Hu et al., 2023) | Det&Map | 97.8 | 91.9 | 92.9 | **100** | 78.8 | 83.4 |
| PARA-Drive (Weng et al., 2024) | Det&Map | 97.9 | 92.4 | 93.0 | 99.8 | 79.3 | 84.0 |
| Transfuser (Chitta et al., 2022) | Det&Map | 97.7 | 92.8 | 92.8 | **100** | 79.2 | 84.0 |
| DRAMA (Yuan et al., 2024a) | Det&Map | 98.0 | 93.1 | **94.8** | **100** | 80.1 | 85.5 |
| VADv2 (Chen et al., 2024) | Det&Map | 97.2 | 89.1 | 91.6 | **100** | 76.0 | 80.9 |
| Hydra-MDP-W-EP (Li et al., 2024b) | Det&Map | **98.3** | 96.0 | 94.6 | **100** | 78.7 | 86.5 |
| DiffusinDrive (Liao et al., 2025) | Det&Map | 98.2 | 96.2 | 94.7 | **100** | 82.2 | 88.1 |
| ResWorld | Det&Map | 98.2 | 96.4 | 98.9 | **100** | 82.5 | 88.3 |
| ResWorld$^\star$ | Det&Map | 98.9 | **96.5** | 95.6 | **100** | **83.1** | **89.0** |

planning model. We adopt the evaluation methods of both UniAD (Hu et al., 2023) and VAD (Jiang et al., 2023) to compare with as many methods as possible, where VAD's metrics are the temporal averages of UniAD's metrics.

**NAVSIM** We further conduct the closed-loop evaluation of ResWorld on the NAVSIM benchmark (Dauner et al., 2024). The data of NAVSIM benchmark are resampled from OpenScene (Contributors, 2023), which contains 120 hours of driving logs selected from the nuPlan dataset (Caesar et al., 2021). By removing simple scenarios from OpenScene, such as straight-driving scenarios, the evaluative capability of NAVSIM benchmark for planning models is enhanced. NAVSIM benchmark employs the Predictive Driver Model Score (PDMS) to comprehensively evaluate the planning model, which is calculated using five key factors including No At-Fault Collision (NC), Drivable Area Compliance (DAC), Time-to-Collision (TTC), Comfort (Comf.), and Ego Progress (EP).

### 4.2 IMPLEMENTATION DETAILS

**nuScenes** When conducting experiments on the nuScenes benchmark, we adopt a model structure similar to SSR (Li & Cui, 2025). To extract the temporal residuals of BEV features, we replaced the BEVFormer (Li et al., 2022b) used in SSR with GeoBEV (Zhang et al., 2025), aiming to generate high geometric quality BEV features at different timestamps separately. We adopt ResNet-50 (He et al., 2016) as the image backbone to process the multi-view images downsampled to $256 \times 704$. We set $k = 2$, which means data from the current frame and 2 previous frames are used. It is optional to use ego status in the planning module, which corresponds to the "in Planner" configuration in BEVPlanner (Li et al., 2024c). Metrics for both configurations are reported. The model is trained for 12 epochs on 8 NVIDIA RTX 3090 GPUs with a total batch size of 8. The AdamW (Loshchilov & Hutter, 2019) optimizer with a learning rate of $1 \times 10^{-4}$ is utilized. We further conduct ablation studies on the nuScenes benchmark to evaluate the effectiveness of the proposed components.

**NAVSIM** For experiments conducted on NAVSIM benchmark, we adopt a model structure similar to TransFuser (Chitta et al., 2022), which utilized two ResNet-34 backbones to process concatenated images and LiDAR BEV maps. Since previous methods did not utilize historical frames, we used the agent queries employed in object detection to replace temporal residuals as the input of the world model. We also implemented a version without auxiliary tasks, which is used for comparison with perception-free models. The model is trained for 100 epochs on 8 NVIDIA RTX 3090 GPUs with a total batch size of 512 and the learning rate is set to $6 \times 10^{-4}$.

### 4.3 MAIN RESULTS

**nuScenes** We conducted comprehensive comparisons between ResWorld and existing end-to-end autonomous driving methods on the nuScenes (Caesar et al., 2020) benchmark as shown in Tab 1. It can be found that ResWorld achieved a new state-of-the-art accuracy. When ego status is not used in the planning module, our method outperforms methods such as GenAD (Zheng et al., 2024b) and

Table 3: **Ablation study of each proposed component.** "TR-World" and FGTR denote Temporal Residual World Model and the Future-Guided Trajectory Refinement, respectively.

| Ego Status in Planner | TR-World | FGTR | L2 (m) ↓ | | | | Collision Rate (%) ↓ | | | |
|---|---|---|---|---|---|---|---|---|---|---|
| | | | 1s | 2s | 3s | Avg | 1s | 2s | 3s | Avg |
| ✗ | | | 0.25 | 0.62 | 1.27 | 0.71 | **0.02** | 0.25 | 0.64 | 0.31 |
| ✗ | ✓ | ✓ | 0.22 | 0.56 | 1.17 | 0.65 | **0.02** | **0.04** | 0.64 | 0.23 |
| ✓ | | | 0.21 | 0.55 | 1.18 | 0.65 | **0.02** | 0.12 | 0.70 | 0.28 |
| ✓ | ✓ | | **0.19** | 0.51 | 1.12 | 0.61 | **0.02** | 0.10 | 0.64 | 0.25 |
| ✓ | | ✓ | 0.20 | 0.52 | 1.12 | 0.61 | **0.02** | 0.10 | 0.55 | 0.22 |
| ✓ | ✓ | ✓ | **0.19** | **0.50** | **1.08** | **0.59** | **0.02** | 0.06 | **0.43** | **0.17** |

Table 4: **Ablation study of Temporal Residual World Model.** "Future Supervision" denotes the utilization of real future data to supervise the future BEV features predicted by the world model.

| World Model Type | Future Supervision | L2 (m) ↓ | | | | Collision Rate (%) ↓ | | | |
|---|---|---|---|---|---|---|---|---|---|
| | | 1s | 2s | 3s | Avg | 1s | 2s | 3s | Avg |
| Normal World Model | ✓ | 0.20 | 0.52 | 1.11 | 0.61 | **0.02** | 0.12 | 0.57 | 0.23 |
| | ✗ | 0.20 | 0.53 | 1.11 | 0.61 | **0.02** | 0.12 | 0.49 | 0.21 |
| TR-World | ✓ | **0.19** | 0.51 | 1.12 | 0.61 | **0.02** | 0.08 | 0.53 | 0.21 |
| | ✗ | **0.19** | **0.50** | **1.08** | **0.59** | **0.02** | **0.06** | **0.43** | **0.17** |

DiffusionDrive (Liao et al., 2025), which rely on auxiliary perception/prediction tasks for driving scene understanding. ResWorld also demonstrates advantages over methods like SSR (Li & Cui, 2025) and LAW (Li et al., 2025), which employ none of the auxiliary tasks and fully rely on world models for scene understanding. When adopting ego status to help predict more accurate prior trajectories, the final planning accuracy of ResWorld is largely improved and outperforms BEV-Planner++ (Li et al., 2024c). This indicates that our framework boasts robust scene understanding capability, which prevents overfitting due to over-reliance on ego status.

**NAVSIM** We also evaluated the closed-loop planning accuracy of ResWorld on NAVSIM (Dauner et al., 2024) benchmark, and the experimental results are presented in Tab 2. To conduct a fair comparison with methods that do not utilize historical frame data, agent queries for object detection replace temporal residuals as the input of TR-World. Nevertheless, ResWorld still achieves the state-of-the-art planning accuracy of 88.3% PDMS, surpassing the performance of Hydra-MDP (Li et al., 2024b) and DiffusionDrive (Liao et al., 2025). When not using auxiliary tasks such as detection and BEV map segmentation, our proposed method also outperforms world model-based methods like LAW (Li et al., 2025) and World4Drive (Zheng et al., 2025). Furthermore, the complete ResWorld implemented with historical frame data achieves 89.0% PDMS, demonstrating the capacity of temporal residuals to represent dynamic information of the driving scene.

## 4.4 ABLATION STUDY

**Efficiency of Components** We conducted experiments to evaluate the effectiveness of Temporal Residual World Model (TR-World) and Future-Guided Trajectory Refinement (FGTR) module, and the experimental results are shown in Tab 3. When only using TR-World and implicitly optimizing trajectories in the manner of SSR, it can significantly improve the model's scene understanding capability and enhance planning accuracy. When only using the FGTR module and refining prior trajectories with the current BEV features, it can also effectively improve the quality of trajectories. The combination of TR-World and FGTR can further improve the planning performance. When not using ego status in the planning module, the two modules together reduce 8.4% of the baseline's average L2 error and 25.8% of the baseline's average collision rate. When adopting ego status in the planning module, the two modules also reduce 9.2% of the baseline's average L2 error and 39.3% of the baseline's average collision rate.

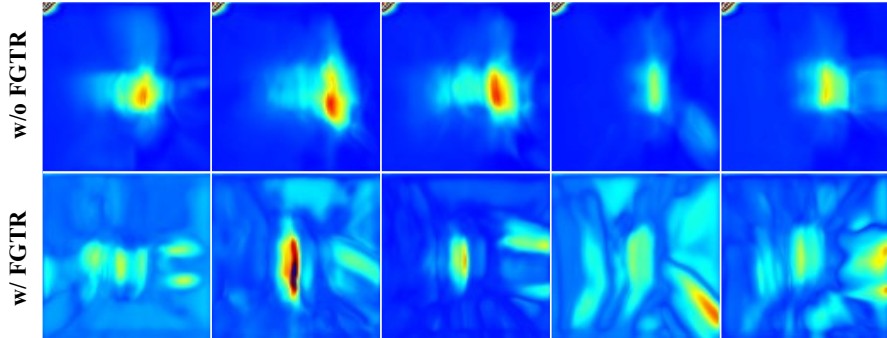

Figure 4: **Effect of Future-Guided Trajectory Refinement Module on alleviating world model collapse.** The first row presents the future BEV features supervised using real future data, while those in the second row are predicted by the world model equipped with FGTR module. The BEV features in the second row show more diversity in spatial distribution.

**Temporal Residual World Model** In Tab 4, we compare the performance of TR-World and the normal world model. The impact of using real future data to supervise the prediction of the world model is also evaluated. It can be found that TR-World, which takes temporal residuals as input and focuses on dynamic object modeling, can predict more accurate future BEV features than the normal world model, thereby achieving higher planning accuracy. Furthermore, the sparse spatial-temporal supervision effect of FGTR module enables TR-World to predict scene information for a future time period, instead of being limited to the scene at timestamp t+1. Therefore, if the data at time t+1 is used for future supervision, it will instead cause the future BEV representation to lose richer temporal information, leading to a decrease in planning performance. In contrast, the normal world model devotes most of its efforts to redundant static object modeling, leading to less accurate predictions of dynamic objects. This explains why future supervision has little impact on the performance of the normal world model.

**Future-Guided Trajectory Refinement** To verify the impact of FGTR module in alleviating world model collapse, we visualize the future BEV features predicted by the world model and present them in Fig 4. It can be observed that for the world model without FGTR module, the predicted future BEV features across different driving scenes show little difference and fail to exhibit complete spatial information. In contrast, through the interaction between prior trajectories and the predicted future BEV features at specific spatial points, FGTR module can urge the world model to predict accurate spatial information, thereby effectively preventing the world model from collapsing.

**Performance of Prior Trajectory** We also evaluate the metric of the prior trajectory and show the results in Tab 5. It can be observed that although the model structure used for generating prior trajectories is the same as the baseline, the prior trajectories have achieved a significant accuracy improvement compared with the baseline. This is because BEV features of the scene are effectively optimized by TR-World and FGTR modules, thereby enhancing the planning capability of the base model. This also proposes a new approach of utilizing larger-scale TR-World and FGTR modules during training to obtain the best BEV features, while taking prior trajectories as output for higher efficiency during inference.

Table 5: **Performance of Prior Performance.** The prior trajectory is predicted using the same model architecture as that of the baseline, while the prediction of the final trajectory requires the TR-World and FGTR models.

| Trajectory | L2 (m) ↓ | | | | Collision Rate (%) ↓ | | | |
|---|---|---|---|---|---|---|---|---|
| | 1s | 2s | 3s | Avg | 1s | 2s | 3s | Avg |
| Baseline | 0.21 | 0.55 | 1.18 | 0.65 | **0.02** | 0.12 | 0.70 | 0.28 |
| Prior Trajectory | 0.20 | 0.53 | 1.12 | 0.61 | **0.02** | 0.10 | **0.41** | 0.18 |
| Final Trajectory | **0.19** | **0.50** | **1.08** | **0.59** | **0.02** | **0.06** | 0.43 | **0.17** |

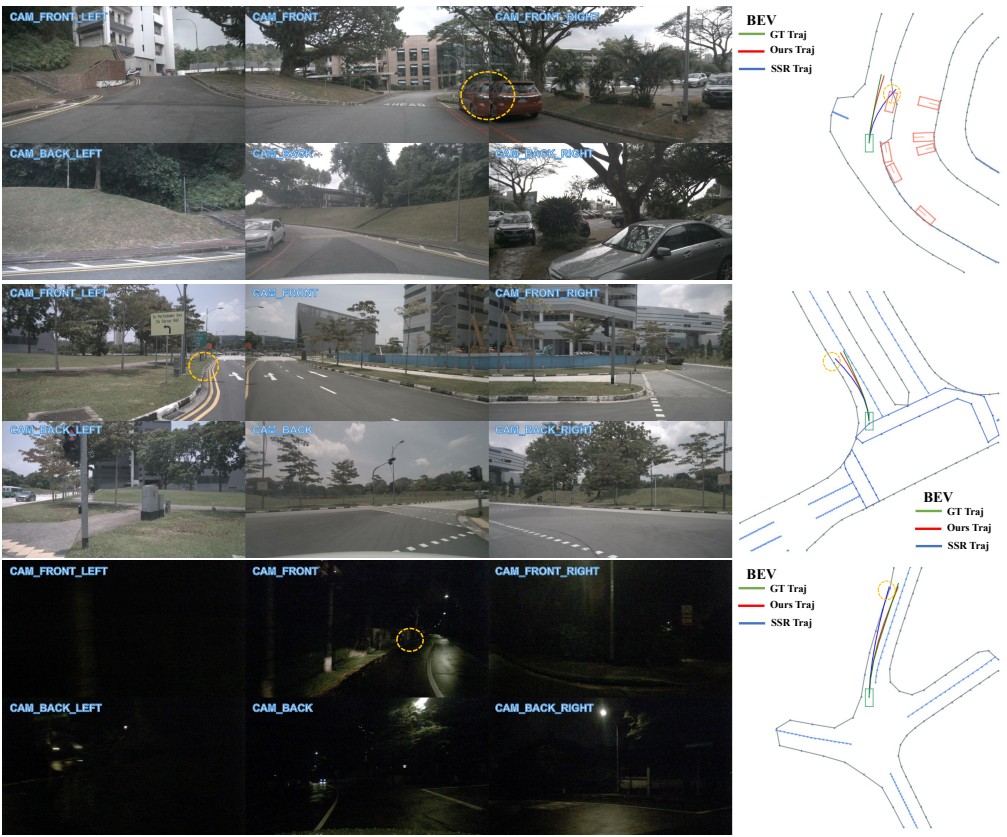

Figure 5: **Visualization of Planning Results**. The object bounding boxes and lane lines on the BEV plane are rendered using the annotations. The green box denotes the ego vehicle. The areas enclosed by dashed circles indicate where collisions will occur.

## 4.5 VISUALIZATION

We compare the qualitative results of ResWorld with SSR (Li & Cui, 2025) on planning trajectories in Fig 5. It can be observed that the trajectories predicted by our method can effectively avoid collisions with other vehicles or curbs, demonstrating a stronger scene understanding ability.

## 5 CONCLUSION

Our proposed ResWorld adopts a novel Temporal Residual World Model framework. It captures information about dynamic objects by calculating the temporal residuals of scene representations. This allows the world model driven by temporal residual to focus explicitly on forecasting the future spatial distribution of dynamic objects, eliminating the redundant modeling of static objects. Furthermore, through the Future-Guided Trajectory Refinement module, the predicted future BEV features are utilized to correct prior trajectories, thereby reducing the probability of driving accidents. The spatial-temporal interaction between prior trajectories and future BEV features also serves as a sparse supervision on the latent world model, effectively alleviating the model collapse. ResWorld achieves state-of-the-art planning performance on both nuScenes and NAVSIM benchmarks.

**Limitations and Future Work** While TR-World demonstrates greater sensitivity to subtle movements of dynamic objects than existing world models (e.g., SSR, LAW), it cannot adequately capture potential dynamic objects (e.g. pedestrians and parked cars) through temporal residuals. As a result, such objects can only be processed alongside static objects by the prior trajectory prediction branch. Our future work will focus on how to use coarse perception to extract the information of potential dynamic objects from the scene and perform preventive modeling for them. This will further enhance the safety of the planning results predicted by our framework.

ACKNOWLEDGMENTS

This research was supported by Zhejiang Provincial Natural Science Foundation of China under Grant No. LD24F020016 and National Natural Science Foundation of China under Grant No. 62576023.

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
