# OpenReview forum: "ResWorld: Temporal Residual World Model for End-to-End Autonomous Driving"
_ICLR.cc/2026/Conference — ICLR 2026 Poster_

### Official Review · Reviewer_podK · 2025-10-22

**Soundness:** 3
**Presentation:** 3
**Contribution:** 2
**Rating:** 4
**Confidence:** 5

**Summary:**

This paper proposes a residual-based world model for end-to-end autonomous driving planning. Unlike previous works that predict future ego-centric image frames or BEV features, this approach centers on the current time step and leverages the differences among multiple BEV feature frames to make the model focus more on dynamic object motion while reducing redundant predictions of static elements. A deformable attention mechanism is then used to interact the prior trajectory with the predicted future BEV features to generate the final planning trajectory. The method demonstrates noticeable improvements on the nuScenes and NAVSIM datasets.

**Strengths:**

1. The idea of utilizing the static information from the current frame to avoid redundant predictions of static objects in future frames, thereby focusing more on dynamic targets, is very insightful.
2. On the recent popular NAVSIM benchmark, the proposed method achieves a noticeable improvement in planning performance compared to previous approaches.

**Weaknesses:**

1. The statements in the abstract, line 101, and the experimental tables suggesting that the proposed method does not rely on auxiliary tasks are somewhat misleading, since the BEV encoder still requires supervision from detection and mapping tasks during training.
2. A major weakness of the paper is the lack of evaluation on inference speed. Given that the proposed method introduces multiple designs for feature interaction and adopts a two-stage trajectory generation process, it is unclear whether these additions incur significant computational overhead.
3. Including ablation studies on the NAVSIM dataset would make the results more convincing. The nuScenes dataset contains a large proportion of straight-driving scenarios, and its metrics are already close to saturation, making it difficult to determine whether the reported improvements come from the proposed design or just statistical variation.

**Questions:**

In line 269, the paper states that “Experiments confirm that not supervising $B_{future}$ enables higher planning performance.” As I understand it, $B_{future}$ is obtained by fusing the current BEV features with the temporally redundant dynamic features from historical frames. It is therefore unclear why $B_{future}$ can “preserve the spatial distribution of dynamic objects across multiple future timestamps.” In my view, since end-to-end autonomous driving is a multi-task framework where different tasks can influence each other, using the ground truth BEV features as supervision provides a very dense supervisory signal, which may actually interfere with the planning performance. Therefore, I doubt the claimed reason may not be the true cause of the observed improvement.

---

> ### Author Response · Authors · 2025-11-21
> **Thanks and response to Reviewer podK**
>
> Thank you for your valuable comments and constructive feedback. We address each of your points below:
>
> **Response to W1: Auxiliary tasks**
> Except for experiments on NAVSIM benchmark shown in Table 2, we do not use auxiliary tasks such as detection and map prediction in the training process of ResWorld. On NAVSIM, we used auxiliary tasks primarily to ensure a fair comparison with methods that also employ auxiliary tasks, such as Hydra-MDP and DiffusinDrive. The version of ResWorld without auxiliary tasks also outperforms other methods that do not use auxiliary tasks in planning performance, such as LAW and World4Drive.
>
> ---
>
> **Response to W2: Evaluation on inference speed**
> We evaluate the inference speed of ResWorld on a single RTX 3090 GPU. The results are as follows:
> | Trajectory | L2 1s | L2 2s | L2 3s | L2 Avg | CR 1s | CR 2s | CR 3s | CR Avg | FPS |
> | --- | --- | --- | --- | --- | --- | --- | --- | --- | --- |
> | Baseline | 0.21 | 0.55 | 1.18 | 0.65 | 0.02 | 0.12 | 0.70 | 0.28 | 21.0 |
> | Prior Trajectory | 0.20 | 0.53 | 1.12 | 0.61 | 0.02 | 0.10 | 0.41 | 0.18 | 21.0 |
> | Final Trajectory | 0.19 | 0.50 | 1.08 | 0.59 | 0.02 | 0.06 | 0.43 | 0.17 | 17.7 |
>
> The complete ResWorld achieves 17.7 FPS, showing good real-time performance. In addition, TR-World and FGTR modules effectively enhance the ability of BEV features to represent information of driving scenes. Even without using the TR-World and FGTR modules for inference, the predicted prior trajectories still show a significant improvement over the baseline. At this point, the model can achieve 21.0 FPS.
>
> ---
>
> **Response to W3: Ablation study on NAVSIM**
> We conduct an ablation study on NAVSIM to verify the effectiveness of TR-World and FGTR modules. Besides, we also supplement the experiment that using history frames to fully evaluate the dynamic modeling capability of TR-World. The experiment results are as follows:
>
> | TR-World | FGTR | History Frame | NC | DAC | TTC | Comf. | EP | PDMS |
> | --- | --- | --- | --- | --- | --- | --- | --- | --- |
> | ✗ | ✗ | ✗ | 98.1 | 95.5 | 94.0 | 100 | 81.7 | 87.1 |
> | ✗ | ✓ | ✗ | 98.3 | 96.1 | 94.8 | 100 | 82.4 | 88.1 |
> | ✓ | ✓ | ✗ | 98.2 | 96.4 | 94.8 | 100 | 82.5 | 88.3 |
> | ✓ | ✓ | ✓ | 98.9 | 96.5 | 95.6 | 100 | 83.1 | 89.0 |
>
> It can be observed that the FGTR module has improved the PDMS score of the baseline by 1.0, while TR-World that uses agent queries instead of temporal residuals partially improves the model accuracy. After using history frames to calculate real temporal residuals as input, the PDMS score is further improved by 0.7. This demonstrates the effectiveness of TR-World and the FGTR module in complicated driving scenes.
>
> ---
>
> **Response to Q1: Future supervision**
> In Table 4, we applied future supervision to both the normal world model (which takes scene queries as input) and TR-World (which takes temporal residuals as input). Despite TR-World’s stronger capability in dynamic object modeling and prediction, their accuracy differs slightly after incorporating future supervision. This is because future supervision aligns the predictions of both the normal world model and TR-World with the future BEV feature at timestamp t+1. Since static objects account for a larger proportion in BEV features, the dense future supervision prevents TR-World from leveraging its advantages in dynamic object modeling.
> After removing future supervision, the sparse spatial-temporal supervision implemented by FGTR dominates the optimization direction. Both the normal world model and TR-World are encouraged to predict cross-temporal dynamic object information that is beneficial for long-term planning. Consequently, TR-World’s advantage in dynamic object modeling is highlighted, thus enabling it to outperform the normal world model in planning performance.

---

> > ### Comment · Reviewer_podK · 2025-11-28
> >
> > Thanks for the detailed clarification, most of my concerns have been addressed.

---

### Official Review · Reviewer_PiGQ · 2025-10-28

**Soundness:** 2
**Presentation:** 2
**Contribution:** 2
**Rating:** 4
**Confidence:** 4

**Summary:**

This paper introduces ResWorld, an end-to-end autonomous driving system centering on a Temporal Residual World Model that selectively models dynamic objects via temporal residuals in BEV representations, reducing redundancy and improving dynamic prediction. The system further proposes a Future-Guided Trajectory Refinement mechanism that explicitly interacts predicted trajectories with future BEV features, aiming to improve planning accuracy and avoid “world model collapse.” Experiments on nuScenes and NAVSIM benchmarks show that ResWorld achieves superior performance.

**Strengths:**

- Temporal residuals in BEV separate dynamic from static content, alleviating redundant static region modeling and weak interaction between the world model and trajectories.
- FGTR explicitly couples predicted trajectories with future BEV features under supervision, improving robustness.
- Experiments on Nuscenes and NavSim demonstrate the superiority of the proposed methods.

**Weaknesses:**

- ResWorld utilizes GeoBEV to generate high-quality BEV features. Compared to other methods, especially the baseline SSR, this approach may be considered unfair. Moreover, the resolution of the BEV features and other hyperparameters, such as the number and dimensions of scene tokens, are not clearly defined, making it challenging to practically verify the effectiveness of the proposed method.
- The analysis and conclusions presented in Table 4 are both confusing and overstated, lacking sufficient experimental evidence. For example, the results in Table 4 mainly show that the impact of future supervision on planning is minimal. This makes it difficult to conclude that TR-World focuses more on dynamic targets or produces more accurate BEV features. Furthermore, the claim that TR-World retains more detailed information about the spatial distribution of dynamic targets is not adequately supported by experimental data.
- There is a lack of experiments on a closed-loop benchmark.

**Questions:**

- Why do the collision rate results of SSR reported in this paper differ so significantly from those in the original publication?

---

> ### Author Response · Authors · 2025-11-21
> **Thanks and response to Reviewer PiGQ**
>
> Thank you for your valuable comments and constructive feedback. We address each of your points below:
>
> **Response to W1: SSR implemeted with GeoBEV**
> We implement SSR using GeoBEV and conduct a fair comparison with ResWorld. The results in UniAD-style are as follows:
>
> | Method | L2 1s | L2 2s | L2 3s | L2 Avg | CR 1s | CR 2s | CR 3s | CR Avg |
> | --- | --- | --- | --- | --- | --- | --- | --- | --- |
> | SSR | 0.25 | 0.64 | 1.33 | 0.74 | 0.08  | 0.12 | 0.72 | 0.31 |
> | SSR (GeoBEV) | 0.25 | 0.61 | 1.26 | 0.71 | 0.02  | 0.12 | 0.63 | 0.25 |
> | ResWorld | 0.22 | 0.56 | 1.17 | 0.65 | 0.02  | 0.04 | 0.64 | 0.23 |
>
> Both SSR (GeoBEV) and ResWorld utilize Tokenlearner module to convert BEV features with a resolution of 100 $\times$ 100 into 16 scene queries. It can be observed that higher-quality BEV features generated by GeoBEV can also improve the planning capability of SSR. Meanwhile, ResWorld still maintains a distinct advantage in planning performance, which demonstrates the effectiveness of our proposed TR-World and FGTR.
>
>
> ---
>
> **Response to W2: Explanation of Table 4**
> We apologize for not clearly explaining the meaning of each configuration in Table 4. We present Table 4 below for detailed explanation.
>
> | World Model Type | Future Supervision | L2 1s | L2 2s | L2 3s | L2 Avg | CR 1s | CR 2s | CR 3s | CR Avg |
> | --- | --- | --- | --- | --- | --- | --- | --- | --- | --- |
> | Normal World Model | ✓ | 0.20 | 0.52 | 1.11 | 0.61 | 0.02 | 0.12 | 0.57 | 0.23 |
> | Normal World Model | ✗ | 0.20 | 0.53 | 1.11 | 0.61 | 0.02 | 0.12 | 0.49 | 0.21 |
> | TR-World | ✓ | 0.19 | 0.51 | 1.12 | 0.61 | 0.02 | 0.08 | 0.53 | 0.21 |
> | TR-World | ✗ | 0.19 | 0.50 | 1.08 | 0.59 | 0.02 | 0.06 | 0.43 | 0.17|
>
> We compare the accuracy performance of the normal world model (which takes scene queries as input) and TR-World (which takes temporal residuals as input), and separately demonstrate the impact of future supervision on both. All configurations utilize the FGTR module.
> For TR-World, removing future supervision can instead improve the model's planning accuracy. This is because the FGTR module already serves the function of sparse spatial-temporal supervision, enabling future BEV features to adaptively express cross-temporal information of the future scene. Adding future supervision would restrict the information of future BEV features to the T+1 timestamp, thereby losing information useful for longer-term planning.
> When not using future supervision, TR-World achieves better planning performance than the normal world model. This indicates that TR-World, which takes temporal residuals as input and focuses on dynamic object modeling, can retain richer cross-temporal spatial information in the future BEV features. In contrast, the normal world model has to allocate part of its prediction capability to redundant static object modeling, thereby impairing the prediction quality of dynamic object information.
>
> ---
>
> **Response to W3: Experiments on close-loop benchmark**
>
> We have conducted experiments on ResWorld using NAVSIM benchmark and achieved SOTA performance. NAVSIM, as a pseudo closed-loop benchmark, also evaluates closed-loop metrics of the planning results, including No At-Fault Collision (NC), Drivable Area Compliance (DAC), Time-to-Collision (TTC), Comfort (Comf.) and and Ego Progress (EP). Unlike simple open-loop metrics such as L2 loss, these closed-loop metrics can comprehensively evaluate the model’s planning capability. In addition, NAVSIM consists of complex driving scenes selected from massive real-world data instead of simple straight-driving scenes, making it more capable of reflecting the planning model’s ability to handle complex road conditions.
> To verify the performance of the complete ResWorld on NAVSIM, we used history frames to calculate real temporal residuals for TR-World, and the results show that the PDMS score of ResWorld is further improved by 0.7.
> | Method | History Frame | NC | DAC | TTC | Comf. | EP | PDMS |
> | --- | --- | --- | --- | --- | --- | --- | --- |
> | ResWorld | ✗ | 98.2 | 96.4 | 94.8 | 100 | 82.5 | 88.3 |
> | ResWorld | ✓ | 98.9 | 96.5 | 95.6 | 100 | 83.1 | 89.0 |
>
> We also plan to evaluate our method on more closed-loop benchmarks, such as Bench2Drive. We will update the results once the experiments are completed.
>
>
> ---
>
> **Response to Q1: Collision rate of SSR**
> An [issue](https://github.com/PeidongLi/SSR/issues/14) in the official repository of SSR shows that many researchers have reproduced the same collision rate using the officially provided model weights, which differs from the collision rate reported in the SSR paper. Therefore, we cite the reproduced collision rate in Table 1.

---

> > ### Comment · Reviewer_PiGQ · 2025-11-26
> >
> > Thank you for the detailed rebuttal and for clarifying the implementation details (e.g., SSR with GeoBEV), the configurations in Table 4, and the use of NAVSIM.
> >
> > While these explanations help me better understand the method, I still feel that the practical gains are relatively modest and somewhat incremental, especially given the added complexity. Some claims about TR-World's focus on dynamic objects and the effect of future supervision also seem stronger than what the current evidence supports, as the experimental improvements appear quite small.
> >
> > If possible, it would be beneficial to strengthen further the closed-loop evaluation on Bench2Drive or the Town05 Long benchmark, in order to provide a fair comparison with the SSR baseline.
> >
> > I appreciate the additional efforts and clarifications; however, given the current version of the paper and the presented results, I am unfortunately not in a position to raise my scores.

---

> ### Author Response · Authors · 2025-11-27
> **Thanks for the feedback**
>
> Thank you again for taking the time to review our response. Here, we would like to provide additional explanations regarding your consideration.
>
> **1. The Future Prospects of ResWorld in the World Model**
> We believe that ResWorld represents not merely an incremental advancement over SSR, but rather a large innovation in the entire field of world model. The characteristics of TR-World in avoiding static redundant modeling and the posterior correction capability of the FGTR module can be applied to various applications, including autonomous driving, robotics, and virtual simulation.
> Taking the robotics field as an example, by constructing a 3D representation of indoor scenes under a unified coordinate system, TR-world can free robots from the cumbersome and redundant static object modeling. The temporal residuals can quickly capture valuable information in the indoor scenes (e.g., human/pet movements or previously occluded objects that have just been discovered), enabling robots to make timely and appropriate responses more effectively. FGTR can endow robots with the capability of posterior correction for actions. For instance, the future 3D coordinates of the robot’s joint points can be calculated from the predicted actions, which can serve as anchors to extract information from the future 3D scene representation. The extracted information can be used for the posterior correction of the action, effectively avoiding undesirable actions (e.g., collisions) or generating more reasonable actions (e.g., grasping a cup more smoothly). In addition, the spatial-temporal joint supervision function of FGTR can effectively prevent world model collapse and enhance the stability of the self-supervised framework, facilitating the effective utilization of large-scale unlabeled data.
> We hope that you could help amplify the influence of ResWorld, promoting the industrial deployment of world models in autonomous driving, robotics, and other applications. We believe ResWorld’s potential can be fully demonstrated by leveraging large-scale data and verification platforms from the industry.
>
> **2. Improvements Over SSR**
> Since the nuScenes dataset contains a large number of simple straight-driving scenes, it dilutes the improvement of ResWorld’s planning capability over SSR in complicated road conditions, such as turning scenes. Therefore, we separately calculate the L2 loss for straight-driving and turning scenes, and the results are as follows:
> *L2 Loss of straight-driving scenes*
> | Method | L2 1s | L2 2s | L2 3s | L2 Avg |
> | --- | --- | --- | --- | --- |
> | SSR (GeoBEV) | 0.24 | 0.56 | 1.17 | 0.66 |
> | ResWorld | 0.21 | 0.52 | 1.10 | 0.61  |
>
> *L2 Loss of turing scenes*
> | Method | L2 1s | L2 2s | L2 3s | L2 Avg |
> | --- | --- | --- | --- | --- |
> | SSR (GeoBEV) | 0.36 | 0.94 | 1.90 | 1.07 |
> | ResWorld | 0.32 | 0.80 | 1.59 | 0.90 |
>
> It can be observed that ResWorld achieves a more significant improvement over SSR in turning scenes. Particularly, L2 loss at 3s is reduced by 0.41m, which is a substantial optimization in practical driving scenes. The visualization results presented in Fig 5 and 6 also confirm the significant improvement of ResWorld over SSR in turning scenes.
> We are currently conducting closed-loop evaluations of ResWorld on Bench2Drive and Town05 Long benchmarks. To quickly compare the closed-loop metrics of ResWorld and SSR, we first reproduce SSR on NAVSIM under the same experimental settings as ResWorld. The results further demonstrate the advantage of ResWorld in driving capabilities.
> | Method | History Frame | NC | DAC | TTC | Comf. | EP | PDMS |
> | --- | --- | --- | --- | --- | --- | --- | --- |
> | SSR | ✓ | 98.6 | 95.8 | 95.1 | 100 | 82.5 | 88.2 |
> | ResWorld | ✓ | 98.9 | 96.5 | 95.6 | 100 | 83.1 | 89.0 |

---

### Official Review · Reviewer_xmSF · 2025-11-01

**Soundness:** 3
**Presentation:** 3
**Contribution:** 2
**Rating:** 6
**Confidence:** 5

**Summary:**

This paper proposes ResWorld, a world model for end-to-end autonomous driving. It presents two main contributions: 1) The Temporal Residual World Model (TR-World), which separates dynamic and static information by calculating the "temporal residual" of BEV features. This allows the world model to focus solely on modeling dynamic objects, avoiding redundant computation on static scenes. 2) The Future-Guided Trajectory Refinement (FGTR) module, which uses the future feature maps predicted by TR-World to explicitly correct and optimize a "prior trajectory," resulting in a safer final plan.

**Strengths:**

1. The FGTR module establishes an explicit feedback loop between the planner and the world model, using predicted future information to correct the current plan, which is logically clear.
2. The method achieves promising results on both the nuScenes (open-loop) and NAVSIM (closed-loop) benchmarks.

**Weaknesses:**

1. A major weakness lies in the experimental evaluation. In the NAVSIM closed-loop tests, the authors admit (Sec 4.2) to not using the core TR-World module, leaving its closed-loop effectiveness unverified.
2. TR-World's residual calculation is highly sensitive to the stability of the BEV features themselves. If the underlying BEV encoder is unstable between frames, its "feature noise" will be conflated with "true motion," leading to an unreliable residual signal.
3. The core assumption of TR-World (residual = dynamic) has a critical safety flaw. As admitted in Appendix B.3, the method fails to model static-but-soon-to-move objects, such as vehicles waiting at a red light.

**Questions:**

1. While the authors state in Sec 4.2 that history frames were omitted for a fair comparison, I believe a closed-loop test of the core innovation (TR-World) is necessary. Can the authors provide closed-loop results for TR-World on NAVSIM to truly validate its contribution?
2. Regarding Weakness 2, how do the authors ensure the learned BEV features are temporally stable? If the features are unstable, how does the model distinguish between residuals from "feature jitter" and residuals from "true object motion"?
3. TR-World treats static-but-soon-to-move objects (like stopped cars and pedestrians) as static background. This is a critical safety failure. How do the authors plan to address this fundamental flaw?
4. How does the model handle future static road structure? If the world model only predicts dynamic residuals and static information comes from the current BEV, how does the vehicle acquire information about the static road ahead when turning or entering an area not covered by the current field of view?
5. The paper argues that the lack of future supervision is an advantage, but this may results in uninterpretability. Could the authors discuss the potential benefits of incorporating explicit future supervision, such as future point cloud prediction [1-2] or future BEV maps/occupancy, to provide a more interpretable signal to the world model?

[1] ViDAR: Visual Point Cloud Forecasting. CVPR 24
[2] HERMES: A Unified Self-Driving World Model for Simultaneous 3D Scene Understanding and Generation. ICCV 25

---

> ### Author Response · Authors · 2025-11-21
> **Thanks and response to Reviewer xmSF**
>
> Thank you for your valuable comments and constructive feedback. We address each of your points below:
>
> **Response to W1 & Q1: Evaluation of TR-World on NAVSIM**
> We supplement the experiment that using history frames in the closed-loop evaluation on NAVSIM to fully evaluate the scene modeling capability of TR-World. We also conducted an ablation study on NAVSIM, and the results are as follows:
>
> | TR-World | FGTR | History Frame | NC | DAC | TTC | Comf. | EP | PDMS |
> | --- | --- | --- | --- | --- | --- | --- | --- | --- |
> | ✗ | ✗ | ✗ | 98.1 | 95.5 | 94.0 | 100 | 81.7 | 87.1 |
> | ✗ | ✓ | ✗ | 98.3 | 96.1 | 94.8 | 100 | 82.4 | 88.1 |
> | ✓ | ✓ | ✗ | 98.2 | 96.4 | 94.8 | 100 | 82.5 | 88.3 |
> | ✓ | ✓ | ✓ | 98.9 | 96.5 | 95.6 | 100 | 83.1 | 89.0 |
>
> It can be observed that the FGTR module has improved the PDMS score of the baseline by 1.0, while TR-World that uses agent queries instead of temporal residuals partially improves the model accuracy. After using history frames to calculate real temporal residuals as input, the PDMS score is further improved by 0.7. This demonstrates the effectiveness of TR-World and the FGTR module in closed-loop evaluation.
>
> ---
>
> **Response to W2 & Q2: Stability of BEV features**
> We load the off-the-shelf GeoBEV, a high-performance BEV detector, to initialize the parameters of ResWorld. Since GeoBEV also fuses BEV features from history frames, it can ensure the spatial consistency of BEV features at different timestamps. Even if there is noise in local regions of the BEV features, the spatial attention map generated using the fused BEV features (integrating cross-temporal information) can function as a filter, mitigating the impact of local noise on scene queries and temporal residuals.
>
> ---
>
> **Response to W3 & Q3: Modeling potential dynamic objects**
> For potential dynamic objects (like stopped cars and pedestrians), ResWorld can mitigate safety risks caused by them in two ways. First, potential dynamic objects are modeled together with other static objects via the prior trajectory prediction branch. Through training on large-scale datasets, the potential future movement range of potential dynamic objects can be summarized, enabling the generation of safer trajectories. Second, increasing the planning frequency (e.g., 10Hz) allows for the timely capture of the motion trends of potential dynamic objects, which are emphasized in temporal residuals for subsequent motion prediction. In fact, TR-World is more sensitive to subtle movements compared to normal world models (e.g., SSR, LAW), thereby effectively enhancing the prediction capability for such objects. In the future, we plan to perform coarse perception on potential dynamic objects. For instance, their information can be captured by predicted masks from images and then integrated into temporal residuals for more accurate predictions.
>
> ---
>
> **Response to Q4: Future static road structure**
> At timestamp t, ResWorld aligns $B_{t-1},\cdots,B_{t-k}$ to the ego-vehicle coordinate system of $B_t$. At timestamp t+1, $B_{t},\cdots,B_{t-k+1}$ are aligned to the ego-vehicle coordinate system of $B_{t+1}$. As a result, static road structures unobservable to the ego-vehicle at timestamp t can be supplemented with images collected at timestamp t+1, enabling the timely updating of trajectories. In addition, increasing the BEV modeling range is an effective approach for the recognition of distant road structures. In the paper, for fair comparison with other methods, ResWorld models road structures within 30m ahead of the vehicle. The modeling range can be extended to around 100m in practical driving applications.
>
> ---
>
> **Response to Q5: Incorporating explicit future supervision**
> Explicit future supervision, which uses future BEV features to predict occupancy or BEV maps of the future driving scenes, can effectively enhance the world model's overall modeling and prediction capabilities, but may lose cross-temporal dynamic information that is essential for planning. Experiment results in Table 4 verify that removing future supervision at specific timestamps applied on TR-World can improve planning accuracy. To incorporate explicit future supervision with TR-World, a feasible approach is to extract spatial information of different future timestamps (e.g., 1s, 2s, 3s) from future BEV features via a set of time-specific embeddings. The time-specific spatial information will then be used to predict occupancy or BEV maps at different timestamps. We plan to refine this approach and verify its effectiveness. Thank you for your valuable insights.

---

### Author Response · Authors · 2025-11-30
**Rebuttal Summary For Area Chair's Convenience**

We provide a concise summary of our rebuttal for the Area Chair’s convenience.

**1. More Evaluation on NAVSIM** (Reviewer xmSF, podK)
In Table 2 of the paper, to fairly compare with other methods, we did not use historical frames when evaluating ResWorld on NAVSIM. Although ResWorld already achieved SOTA accuracy, Reviewer xmSF hopes we can fully evaluate ResWorld on NAVSIM. We supplement this experiment, and the full ResWorld can further increase the PDM score from 88.3 to 89.0.
Based on Reviewer podK's comment, we also conduct ablation studies on NAVSIM, which verify the effectiveness of TR-World and FGTR in closed-loop evaluations. The results are acknowledged by the Reviewer podK.

**2. Modeling Capability and Stability for Driving Scenes** (Reviewer xmSF)
Reviewer xmSF is concerned about ResWorld’s modeling capability for potential dynamic objects and distant road structures.
For potential dynamic objects, their subtle motion trends can be timely captured by temporal residuals, enabling TR-World to achieve faster response capability than normal world models. For distant road structures, they can be continuously updated as the ego vehicle moves forward, enabling a timely response. Expanding the BEV modeling range can also facilitate quickly perception of distant road structures.
Reviewer xmSF also expressed concerns about the temporal stability of the BEV encoder. We explained that ResWorld adopts GeoBEV's parameters for initialization, whose multi-frame fusion capability ensures temporal stability. Deriving scene queries by spatial attention maps also helps filter out local noises.

**3. Future Supervision** (Reviewer xmSF, PiGQ, podK)
Reviewer PiGQ and podK raised concerns about why removing the future supervision on world models can instead improve the ResWorld's prediction for dynamic object information. We explain that future supervision is dense supervision on future scene representation, which causes static objects (with a large proportion in the scene) to dominate the optimization direction. Besides, it only utilizes data at timestamp t+1, which limits the ability of the world model for cross-temporal prediction. FGTR instead achieves the sparse spatial-temporal joint supervision, significantly enhancing the ability to predict the temporal-spatial distribution of dynamic objects. Reviewer PiGQ and podK endorsed our explanation.
Reviewer xmSF was curious about whether ResWorld can be combined with future supervision. We propose an approach that leverages temporal embeddings to decouple the spatial information of different timestamps from future BEV features, with supervision performed using corresponding data. We believe this deserves future exploration.

**4. Comparison with other methods** (Reviewer PiGQ, podK)
Reviewer PiGQ requested the implementation of SSR using GeoBEV to more directly demonstrate the improvements of ResWorld. We conduct this experiment, and the results show that ResWorld still maintains a significant accuracy advantage (especially in turning scenes), which verifies its stronger capability in handling complicated driving scenes.
Reviewer PiGQ also requested a comparison of the closed-loop metrics between ResWorld and SSR. We evaluated SSR on NAVSIM, and the experiments demonstrate that ResWorld also significantly outperforms SSR in closed-loop metrics.
Reviewer podK questioned whether ResWorld used auxiliary tasks like detection or map segmentation. We clarify that auxiliary tasks were only employed when conducting fair comparisons with methods like DiffusionDrive on NAVSIM. Otherwise, ResWorld did not utilize auxiliary tasks.
Reviewer podK was also concerned about the inference speed of ResWorld. Our experiments demonstrate that ResWorld can achieve real-time performance.

**5. Prospects of ResWorld** (Reviewer PiGQ)
In discussion with Reviewer PiGQ, we emphasize the great potential of ResWorld across diverse world model applications, including autonomous driving, robotics, and virtual simulation. Taking robotics as a case study, we propose a comprehensive framework for ResWorld's implementation. We believe both TR-World and FGTR can play a pivotal role in scene understanding and decision optimization for robotics.

We believe that these clarifications effectively address the reviewers’ concerns, and we sincerely appreciate the Area Chair’s time and thoughtful consideration.

---

### Meta-Review · Area_Chair_F78x · 2026-01-08

**Summary:**

Reviewers found the paper to be technically sound and well engineered, with a clear motivation and consistent improvements on nuScenes and NAVSIM. The core idea of focusing world modeling on dynamic objects via temporal BEV residuals was generally viewed as reasonable, and the interaction between future scene prediction and trajectory refinement was appreciated.
However, concerns were raised about the strength and interpretation of the empirical evidence, including whether the reported gains are sufficiently large to support some of the stronger claims, the fairness and clarity of comparisons, and the robustness of the residual-as-dynamic assumption in safety-critical scenarios. These concerns, together with the overall assessment that the contribution is incremental, informed the suggested decision.

**Reviewer Concerns:**

The authors provided additional closed-loop and ablation results on NAVSIM, directly evaluating the core TR-World component and clarifying its contribution; Fairness of comparisons was improved by implementing stronger baselines (e.g., SSR with GeoBEV) and explaining discrepancies with previously reported results. Clarifications were added for experimental settings, inference speed, and auxiliary supervision, resolving several methodological questions raised by reviewers.

Some concerns remain outstanding. The magnitude of the gains remains relatively modest, and some reviewers remain cautious that several claims (e.g., dynamic-object focus and the effect of removing future supervision) are stronger than what the current evidence clearly supports. The residual-equals-dynamic assumption still has limitations in edge cases such as static-but-soon-to-move objects, which are important for safety-critical driving. Broader validation on additional closed-loop benchmarks would further strengthen confidence in the generality of the conclusions.

**Reviewer Scores:**

Reviewer podK (initial score: 4): Most concerns were addressed in the rebuttal, and the reviewer acknowledged this. The score would likely increase slightly, potentially to 6.
Other reviewers may remain their initial scores.

---

### Decision · Program_Chairs · 2026-01-26

Accept (Poster)